# Progress of Polymer-Based Thermally Conductive Materials by Fused Filament Fabrication: A Comprehensive Review

**DOI:** 10.3390/polym14204297

**Published:** 2022-10-13

**Authors:** Zewei Cai, Naveen Thirunavukkarasu, Xuefeng Diao, Haoran Wang, Lixin Wu, Chen Zhang, Jianlei Wang

**Affiliations:** 1College of Chemistry, Fuzhou University, Fuzhou 350116, China; 2CAS Haixi Industrial Technology Innovation Center in Beilun, Ningbo 315830, China; 3CAS Key Laboratory of Design and Assembly of Functional Nanostructures, Fujian Key Laboratory of Nanomaterials, Fujian Institute of Research on the Structure of Matter, Chinese Academy of Sciences, Fuzhou 350002, China; 4Jinyoung (Xiamen) Advanced Materials Technology Co., Ltd., Xiamen 361028, China; 5School of Materials Science and Engineering, Fujian University of Technology, Fuzhou 350118, China; 6School of Materials and Chemistry Engineering, Minjiang University, Xiyuangong Road No. 200, Fuzhou 350108, China; 7Industrial Design Institute, Minjiang University, Xiyuangong Road No. 200, Fuzhou 350108, China

**Keywords:** additive manufacturing, fused filament fabrication, polymer-based thermally conductive material, thermal conductivity

## Abstract

With the miniaturization and integration of electronic products, the heat dissipation efficiency of electronic equipment needs to be further improved. Notably, polymer materials are a choice for electronic equipment matrices because of their advantages of low cost and wide application availability. However, the thermal conductivity of polymers is insufficient to meet heat dissipation requirements, and their improvements remain challenging. For decades, as an efficient manufacturing technology, additive manufacturing has gradually attracted public attention, and researchers have also used this technology to produce new thermally conductive polymer materials. Here, we review the recent research progress of different 3D printing technologies in heat conduction and the thermal conduction mechanism of polymer matrix composites. Based on the classification of fillers, the research progress of thermally conductive materials prepared by fused filament fabrication (FFF) is discussed. It analyzes the internal relationship between FFF process parameters and the thermal conductivity of polymer matrix composites. Finally, this study summarizes the application and future development direction of thermally conductive composites by FFF.

## 1. Introduction

Intelligent electronic devices are currently being researched to meet people’s pursuit of a high-quality life through integration and miniaturization. In order to ensure product safety and operational efficiency, it is imperative to improve the thermal conductivity of electronic devices [1,2]. Polymers are frequently used in preparing heat dissipation materials because of their low price, light weight, ease of processing, and wide applications [3]. However, polymers have a low thermal conductivity, typically 0.1–0.5 W∙m^−1^∙K^−1^ [4,5,6,7], which is insufficient to meet the industrial heat dissipation requirements.

To overcome this problem, high-thermal-conductivity fillers are added to the polymers, such as metal-based fillers, carbon-based fillers, and ceramic-based fillers. It is not only limited to the addition of fillers. Taking maximum advantage of the high thermal conductivity of the filler is also an issue in achieving efficient heat dissipation. So far, many preparation methods have been explored to induce the high orientation of fillers, including freeze casting [8,9,10,11], magnetic induction [5,12,13,14], chemical vapor deposition (CVD) [15], and mechanical force induction [16,17,18]. However, the process strategies have some drawbacks, such as not being easy to prepare, which limits the application of polymers as heat dissipation materials.

The application of additive manufacturing (AM) in polymer-based thermally conductive materials has drawn more attention in recent years. AM technology is also known as the 3D printing technique, which includes fused filament fabrication (FFF), stereolithography apparatus (SLA), direct ink writing (DIW), and so on. The basic principles of these techniques are almost the same, and they can be roughly divided into two processes. First, the model is designed by CAD software, and this model is processed by slicing software and loaded into the printer. Then, the liquid or powder material is bonded layer-by-layer to form the final prints [19]. Moreover, it can produce objects of complex shapes more straightforwardly than conventional manufacturing techniques. Because of these features, it has attracted much interest as a manufacturing technology for thermally conductive materials.

A growing trend has been observed in the literature on thermal conduction research using 3D printing technology since 2000, as shown in Figure 1. The data indicate that researchers have a positive attitude towards the potential of 3D printing in the field of thermal conduction. In particular, FFF has gained wide attention because of its low cost, environmental friendliness, and ease of orientating filler particles. However, due to the stacking principle of the FFF preparation process, there are unavoidable voids between the printed filaments and between the layers leading to heat loss. After adding filler, how to reduce the interfacial thermal resistance between the filler and the substrate still needs further research. In addition, how to produce prints with high thermal conductivity at low filler content is vital to reduce the production cost and minimize the sacrifice of the mechanical properties of the prints. Currently, the heat dissipation of electronic devices has become a problem that can not be ignored, concerning not only the service life of electronic products but also the life safety of users. Taking advantage of FFF’s rapid prototyping and customizability allows for more efficient and tailored production of highly thermally conductive materials. It can be used in small electronic products and is expected to have industrial applications in high-power devices. Therefore, this paper classifies the commonly used thermally conductive fillers in FFF according to their conductive properties and reviews the research progress on preparing polymer-based thermally conductive materials by FFF. The internal relationship between FFF process parameters and the thermal conductivity of polymer matrix composites was analyzed in detail. In addition, common methods for improving the thermal conductivity of FFF thermally conductive materials were demonstrated. This provides a reliable reference for the future investigation of thermally conductive materials by FFF. Finally, it summarized the application fields of 3D printing thermally conductive materials according to the conductive properties of thermally conductive fillers and presented an outlook on future research directions.

## 2. Three-Dimensional Printing in Thermally Conductive Composites

The molding processes of traditional polymer matrix composites include contact molding, compression molding, extrusion molding, and injection molding, all of which require the aid of model-assisted molding without exception, so the mold constrains the part’s shape. The most significant benefit of 3D printing is that it can create prints with intricate shapes without the need for molds, which brings down production costs. The 3D printing process used for molding thermally conductive materials mainly includes FFF, SLA, DIW, and so on [20]. The principles of these techniques and their research progress on thermally conductive composites are briefly introduced in the following.

### 2.1. SLA

The 3D system was founded in 1986 by inventor Charles and entrepreneur Raymond S. Freed, who launched the first SLA printer in 1988, paving the way for commercialization. Consequently, SLA can be regarded as the progenitor of 3D printing. The forming process of SLA is shown in Figure 2. The liquid photopolymer is first added to the resin tank, and the build platform is lowered to a level slightly below the liquid surface of the resin. The laser beam is selectively focused on the surface of the liquid photopolymer to achieve the transition from liquid to solid state. Afterward, the build platform rises to a specific height, and the liquid resin covers the previous print again by laser curing. The resin is stacked layer-by-layer until the printing process is completed. Finally, the finished product is removed from the resin tank for cleaning [21,22,23]. SLA has the benefits of a high level of automation and the ability to make high-resolution prints, so it is frequently employed to produce complicated models. 

The printing material of SLA, also known as photosensitive resin, usually consists of oligomers, reactive diluents, photoinitiators, etc. Oligomers are low-molecular-weight monomers or prepolymers that serve as the photosensitive resin’s main components and determine the printed part’s performance after curing. Choosing oligomers requires consideration of the physical and chemical properties of the cured product on the one hand, and the feasibility of production—such as system viscosity and cost—on the other hand, so there are relatively few options. The most used oligomer in SLA is epoxy acrylate, followed by urethane acrylate. Other oligomers, such as polyester acrylate and polyether acrylate, can also be used as raw materials for photosensitive resins. Active diluents reduce the viscosity of oligomers and accelerate the reaction of resins. The photoinitiator generates reactive intermediates by absorbing radiation energy to activate oligomers and diluent monomers for cross-linking reactions. Since our research on SLA technology is late, the research on photosensitive resin is mainly concentrated in universities and research institutes. As the prints made by SLA have poor mechanical properties and toughness, the researchers used nanoparticles to modify the resin. It can improve the compatibility between the polymer and the filler and further enhance the prints’ performance [24]. Due to the advantages of high precision and rapid prototyping, SLA is regarded as one of the manufacturing technologies of thermal management materials by researchers. In terms of materials, achieving better thermal management starts with designing the cooling structure, including thermal interface materials and packaging materials. In order to dissipate heat without dielectric breakdown, these materials are usually composed of fillers with excellent thermal conductivity and insulation. The most prevalent are alumina [25,26,27,28], silicon carbide [29,30], titania [31,32], etc., which are applied in thermal management materials frequently because of their low price and extensive availability. Azarmi et al. [33] prepared ceramic materials by adding Al_2_O_3_ to a photosensitive polymer resin and tested their thermal properties. After sintering, the porosity of the material decreased from 19.01 ± 1.12% to 8.14 ± 0.85%, while the thermal conductivity showed an increasing trend. The actual measured thermal conductivity increases from 5.17 ± 1.05 W∙m^−1^∙K^−1^ to 26.81 ± 3.5 W∙m^−1^∙K^−1^, confirming that porosity plays a negative role in the heat transfer.

Ceramic fillers with better thermal conductivity, such as boron nitride (BN), aluminum nitride (AIN), and silicon nitride (SIN), have attracted widespread attention. Gurijala et al. [34] made superparamagnetic nanoparticles (SPIONs) lose electrons and bind to negatively charged hBN particles to produce magnetized hBN that low magnetic fields can control (<10 mT). As shown in Figure 3, to solve the problem of difficult magnetic response and orientation of mhBN in the slurry at high filler volume, they added mechanical vibration to reduce the friction between fillers, improve the dispersion of fillers in epoxy resin, and reduce the defects caused by air bubbles. The dual effect of lowering the percolation threshold and minimizing the heat transfer channel is achieved using magnetic-induced fillers orientation. The thermal interface material test of the mhBN-epoxy composites demonstrates that the maximum thermal conductivity can reach 8.67 W∙m^−1^∙K^−1^, which is 36 times higher than that of the pure polymer (0.2 W∙m^−1^∙K^−1^). The combination of magnetic orientation and SLA has produced designable hBN structures, which offer the possibility of achieving intelligent heat transfer in electronic devices. This work presents a new approach for designing thermal management materials required for high-thermal-density applications.

As shown in Figure 4, Mubarak et al. [31] modified Ag on the TiO_2_ surface with the help of the sol–gel method. The obtained Ag-TNP was added to the resin as a filler, which induced photopolymerization of the resin under UV irradiation. Under the influence of ultraviolet light, the nanoparticles will generate valence band holes and conduction band electrons on their surface, which facilitates the formation of monomer radicals, thereby initiating free-radical photopolymerization and enhancing the thermal conductivity and mechanical properties of the polymer. The tensile strength and flexural strength of SLR/Ag-TNP nanocomposites were enhanced with the increase in Ag-TNP content, which were improved by 60.8% and 71.8%, respectively, at a loading of 1.0%. The highest thermal conductivity was achieved at a filler concentration of 1%, with a value of 0.3456 W∙m^−1^∙K^−1^. However, the thermal conductivity showed a decreasing trend with the increase in filler concentration. Therefore, solving the problem of uneven heat dissipation from the polymer matrix caused by the agglomeration phenomenon at high filler content is still a problem that needs further research. The limited availability of raw materials that can be applied to SLA and the significant odor and toxicity of liquid resins have hindered its development to some extent.

### 2.2. DIW

The raw material for DIW is ink with a specific viscosity, usually containing an organic matrix, thickeners, filler particles, and so on. The ink must have both the appropriate shear-thinning rheological behavior to ensure the extrusion process’s smoothness and that the extruded ink can maintain structural stability when in contact with the substrate [35]. DIW works by the three-dimensional movement of the printhead according to the shape of the designed print part, stacked layer by layer to form the initial sample. The curing reaction is then completed under UV light or heat conditions to form the final part. Since the preparation of DIW relies on an extrusion process, the researchers took advantage of this feature to make anisotropic thermally conductive fillers oriented under high shear forces to produce thermally conductive composites.

In most studies, ink extrusion filaments were deposited along the horizontal direction, resulting in two-dimensional fillers oriented along the in-plane direction [36]. As shown in Figure 5, Liu et al. [37] deposited nano-Ag on the BN surface to produce BN-Ag and used them as thermally conductive fillers, where Ag was used as an aid to reduce the thermal resistance between the fillers. The BN-Ag is tightly assembled and aligned along the printing plane due to the shearing during the extrusion process. Therefore, the thermal conductivity of the composite can reach 2.52 W∙m^−1^∙K^−1^ at a low loading of 20 wt%.

When the viscosity of the ink is sufficiently high, direct printing in the vertical direction can be accomplished in addition to layer-by-layer printing in the horizontal direction. As shown in Figure 6, Liang et al. [38] dispersed BN in the F-127 solution, where F-127 was used as a binder to improve the mechanical strength of the printing ink to ensure that the extruded filaments are self-supporting and will not bend or collapse. Due to the anisotropy of the BN shape and the shear force generated when the ink is extruded from the nozzle during printing, the BN nanosheets are tightly packed along the Z-axis to form a continuous thermal conduction path. In order to test the thermal conductivity of the material, they encapsulated a vertical BN-rod array in a polydimethylsiloxane matrix (PDMS) to create a bulk material. The thermal conductivity of the BN-array/PDMS was measured to be 1.50 W∙m^−1^∙K^−1^, whereas the theoretically calculated thermal conductivity of the BN rod is as high as 5.65 W∙m^−1^∙K^−1^. Even though the thermal conductivity of BN-array/PDMS cannot approach the theoretically predicted thermal conductivity, the thermal conductivity of polymer matrix composites can be increased by changing the number of BN rods within the same volume.

The work has apparent advantages, such as the possibility of obtaining a vertically aligned layered structure directly, moving away from the traditional horizontal stack molding. Moreover, the finished product is directly dried overnight without curing post-treatment, providing an avenue for the design of thermal conductivity pathways for the rest of the two-dimensional fillers. However, the disadvantage is also obvious. In order to achieve the purpose of printing ink vertically, the range of the window in which the BN content can be changed is tiny. The extruded ink filament is insufficient to support the molding when the content is less than 50 wt%. The nozzle will be blocked due to excessive ink viscosity when it exceeds 55.6 wt%. Hence, the preparation of vertical printing inks is relatively demanding and challenging. Table 1 lists the thermal conductivity of filled polymers in DIW 3D printing.

### 2.3. FFF

Scott Crump pioneered FFF technology in 1988, which was commercialized by Stratasys the following year, and the first FFF 3D printing machine was sold in 1992. As shown in Figure 7, the working principle is to feed the plastic filament into the gear, throat, and nozzle in sequence. The solid filament is heated by the nozzle to a molten state and deposited on the platform after extrusion. After setting the slicing parameters and generating the g-code format file according to the requirements of the desired part, the nozzle is controlled by the program to move in the X and Y axes. Each time a layer of thickness is deposited, the nozzle is raised upwards by one layer of thickness to form the final printed part [45,46,47,48,49].

FFF has various advantages over other 3D printing technologies: First, FFF printing equipment is inexpensive, but equipment for other printing technologies, such as photocuring and laser sintering, is costly and complicated. Second, the printing material is simple to create, does not have a complex formula design, and the printing process is environmentally friendly and free of pollution. Lastly, the prints produced by FFF can be formed quickly, eliminating the need for lengthy and complex postprocessing for prints that do not require a support structure. In light of these benefits, researchers feel that using FFF 3D printing technology to make polymer-based thermally conductive materials has some practical potential; consequently, several studies have been carried out. The following section briefly introduces the thermal conduction mechanism and then a summary of the literature on FFF thermally conductive materials.

## 3. Thermal Conduction Mechanism

Different materials have various thermal conduction mechanisms, among which solid thermal conduction carriers can be divided into electrons, phonons, and photons. The thermal conduction of metal materials mainly depends on the thermal movement of free electrons, while polymers are usually saturated systems without free electrons; hence, the thermal conduction carrier of polymers is phonon [50]. Phonons are not physical particles, but rather a quantum of energy employed to represent the thermal vibrations of crystal atoms. The crystallinity of the polymer is low, the molecular chains are randomly entangled, and molecular chain vibration will result in phonon scattering. The thermal conductivity of materials is somewhat impacted by phonon scattering [51]. The more frequently a material scatters light, the shorter the phonon mean free path will be, which will result in the material having a weaker thermal conductivity. In 3D printing, thermally conductive fillers are often added to produce filled-type polymers to improve the printed part’s thermal conductivity. The thermal boundary resistance (RB) is caused by the mismatch of phonon frequencies on filler-to-filler or filler-to-matrix, which limits the improvement of the thermal conductivity of filler-type composites [52]. Researchers have proposed several theories better to describe the thermal conductivity behavior of filled polymer composites. The thermal conduction network theory and the percolation thermal conduction theory have the highest acceptance and application [15,53,54,55,56].

### 3.1. Thermal Conduction Network Theory

When the filling amount is small, the filler is dispersed in the matrix in isolation and does not play a significant role in the improvement of the thermal conductivity of the composite. When the filling amount reaches a certain level, the filler and filler begin to connect and form short-range thermal conductivity channels. At this point, the filling amount further increases, and the short-range connected thermal conductivity channels contact each other, forming a wide range of thermal conductivity channels in the matrix. It is difficult for phonons to travel at high speed in polymer chains, so phonons tend to move in the thermal conductivity channels with less resistance, thus achieving the purpose of improving the thermal conductivity of the composite [57]. In this theory, the filler content is an important factor affecting the thermal conductivity of polymer matrix composites. The thermal conductivity network theory is widely used and popular among researchers, especially in explaining anisotropic fillers such as graphene, BN, and graphite flakes. Given this theory, to increase the material’s thermal conductivity, it is only necessary to increase the filler content, which is a relatively simple operation. However, an excessive increase in filler content brings problems such as an increase in production cost and a sacrifice of mechanical properties of prints.

### 3.2. Percolation Thermal Conduction Theory

The theory of conductive percolation refers to the resistivity of the polymer decreasing but remaining insulating when the load is below a threshold value. When the loading exceeds the threshold, the resistivity shows a significant decreasing trend [58]. Some researchers believe that the thermal conductivity of filled polymer composites shows the same trend and therefore explains it by the thermal conductivity percolation theory. Percolation thermal conduction theory is mainly used to explain composites’ thermal conductivity behavior by adding ultrahigh thermal conductivity fillers such as graphene and carbon nanotubes. In this theory, the thermal conductivity of the composite increases significantly when the filler content reaches near the percolation value. As the filler content increases further, the thermal conductivity of most materials tends to increase linearly. Process production guidance can take advantage of this law to minimize production costs while achieving the required thermal conductivity of the product. However, the polymer thermal conductivity does not change nearly as much as the sudden change in resistivity, and its improvement is limited, so the theory is still somewhat controversial.

## 4. Thermally Conductive Fillers in FFF 3D Printing

Polymer thermally conductive materials can be separated into intrinsic and filler types based on their ingredients [59]. Intrinsically thermally conductive materials reduce the resistance to heat transfer by improving the disorderly arrangement of polymer chains through chemical reactions or mechanical action, thereby reducing the entanglement of molecular chains [60,61]. As shown in Figure 8, Xu et al. [62] dissolved semicrystalline polyethylene powder in decalin to make the polymer chains undergo preliminary untwisting. The solution is sent to the Couette flow system, where the shear force during the extrusion process will further reduce the tangling of the molecular chain before the solution flows to the liquid nitrogen-cooled substrate to form a film that will maintain the disentangled structure. As the draw ratio increases, the diameter of the fibers in the film decreases, and the density increases. When the draw ratio reaches 110 times, the polyethylene film is as high as 62 W∙m^−1^∙K^−1^ in the stretching direction, which is higher than many metals and ceramics. Nevertheless, the preparation process for this intrinsic thermal polymer is typically quite complex. In contrast, filler-type thermally conductive polymers are easier to prepare. According to the shape, the thermally conductive fillers can be divided into anisotropic and isotropic fillers. Based on electrical conductivity, they can be classified as insulating and conductive fillers, and different types of fillers can be chosen depending on the application area. Table 2 lists the thermal conductivity of the filler-type polymers prepared using FFF 3D printing.

As shown in Table 2, the FFF thermally conductive fillers mainly include graphene, expandable graphite, carbon fiber, carbon nanotubes, BN, AIN, Cu, Fe, SiC, etc. Expandable graphite can be used as a precursor for preparing graphene, as opposed to researchers who are more interested in using graphene to improve the thermal conductivity of printed parts. Carbon fibers and nanotubes are one-dimensional carbon fiber materials with similar properties, so the more-used carbon fiber was chosen as a representative. Cu and Fe are metal fillers, and both have similar thermal conductivity mechanisms, so Cu, which has more relevant studies and is more representative, was chosen for the introduction. The fillers with similar properties are no longer described in detail. They are divided into two main categories according to the electrical properties of thermally conductive fillers in the following part.

### 4.1. Electrically and Thermally Conductive Fillers

The addition of thermally conductive fillers will not only improve the overall thermal conductivity of the polymer but will also impart the electrical properties of the filler itself to the composite. Commonly used electrically and thermally conductive fillers in FFF include graphene, metal particles, carbon fibers, diamond, graphite, etc. A brief description of these fillers and their research progress in FFF is presented next.

#### 4.1.1. Graphene

Graphene is a material with a monolayer honeycomb lattice structure formed by tightly packed carbon atoms connected by SP^2^ hybridization [83]. As a kind of anisotropic thermally conductive filler, graphene possesses excellent in-plane thermal conductivity (IPTC) of about 2000–5000 W∙m^−1^∙K^−1^ [84], while its through-plane thermal conductivity (TPTC) is only 5–20 W∙m^−1^∙K^−1^ [66]. Due to the vast difference between TPTC and IPTC, maximizing the use of IPTC in thermally conductive composites has become an important issue. In order to maximize the thermal conductivity of the composite, the graphene sheets need to be maximally exfoliated and highly oriented.

However, the typical layered structure of graphene tends to cause folds in the lamellar layers during the compounding process. Moreover, the van der Waals forces between single graphene layers are large, which makes it challenging to achieve exfoliation [85]. In early work, most researchers simply melt-mixed graphene with polymers, thus often resulting in graphene sheet stacks that did not achieve the desired modification. As shown in Figure 9, the composites prepared by Zhu et al. [64] that directly mixed polyamide 12 and GNPs could observe the agglomeration phenomenon (red arrow) under the SEM image, and it became worse as the graphene concentration went up.

Meanwhile, graphene’s degree of dispersion and exfoliation determines whether the thermal conductivity channels can be established most efficiently, dramatically affecting the composite’s thermal conductivity. As a result, the researchers increased the degree of graphene dispersion in the matrix and further exfoliated graphene sheets by combining solution or mechanical mixing with melt mixing. For instance, solid-state shear milling can reduce the number of graphene stacks by repeatedly milling graphene powder and polymer powder [86]. During mechanical mixing, graphene and polymers undergo mechanochemical reactions due to the generation of free radicals [87]. The filler transitions from physical mixing to chemical grafting, which enhances the compatibility between the filler and the polymer matrix, reduce the interfacial thermal resistance and therefore accomplish the goal of enhancing thermal conductivity [66]. Solution mixing here refers to dissolving the polymer in an organic solvent to untwist it, adding graphene powder and stirring to disperse it, and then removing the organic solvent where the polymer improves its dispersion by trapping the graphene and limiting its restacking [68]. The double-mixing method is an excellent way to ensure that graphene is evenly distributed in the polymer matrix and reduces agglomeration.

In addition, the filler’s orientation is crucial for the formation of the thermal transfer network due to the anisotropy of graphene. The researchers believe that most graphene will be horizontally in the X-Y printing plane. In order to take maximum advantage of the IPTC of graphene, the print structure can be designed so that the heat flux direction is aligned with the orientation direction of graphene. By using FFF, it can achieve a higher degree of graphene sheet orientation. However, few researchers have conducted in-depth research on the orientation process. As depicted in Figure 10, Guo et al. [68] thought that the orientation of graphene sheets would occur via the following process: Initially, the graphene would be oriented along the direction of rotation of the twin screw, and then the 1.75 mm filament would be melted and compressed into 0.4 mm microfilament, during which the graphene would change into a vortex morphology protruding from the cross-section. During the printing process, the bottom holds its vortex structure due to the rapid cooling rate, while the top has a certain fluidity due to the relatively slow cooling rate. The compressive effect caused by the deposition process aligns the top graphene sheet horizontally, resulting in asymmetric top and bottom structures.

#### 4.1.2. Cu

Metals have a vast number of electrons that are not constrained, and the interaction or collision that occurs between these electrons can result in the rapid transfer of heat. Therefore, the metal particles are usually added to the polymer to increase the thermal conductivity of the composite material. Among them, copper has become one of the most commonly used fillers to improve the thermal conductivity of polymers due to its low price, abundant reserves, high thermal conductivity, and low coefficient of thermal expansion. The thermal conductivity of copper is 397 W∙m^−1^∙K^−1^, second only to silver, which can achieve excellent results in the heat transfer of composites. However, one of the disadvantages of utilizing copper as a filler is that it is susceptible to oxidation, and the high hardness of copper tends to induce machine wear.

Hwang et al. [81] mixed ABS with copper and iron particles to make a new metal/polymer composite wire that can be used for FFF. When the amount of copper is 50 wt%, the thermal conductivity increases to 0.912 W∙m^−1^∙K^−1^, which is only 41% improved compared with pure ABS (0.646 W∙m^−1^∙K^−1^) and has not improved significantly. As shown in Figure 11, Vu et al. [82] made Cu particles form a separated structure by adding PMMA, which promoted the dispersion of Cu particles and facilitated the phonon transfer in the PLA matrix. The addition of PMMA beads plays a synergistic role in the Cu particles, which is conducive to the formation of the thermal channel of the Cu particles. Following the addition of 20 wt% PMMA beads and 50 wt% Cu particles, the thermal conductivity of the composite material is 317% greater than that of the pure PLA. However, the composites became brittle after the addition of fillers, leading to a decrease in tensile strength and elongation at break, which showed the same trend as the research results of Hwang et al. Therefore, how to ensure the improvement of thermal conductivity without damaging the mechanical properties is also a significant direction to be explored in the future.

#### 4.1.3. Carbon Fiber (CF)

Carbon fibers are composed of incomplete graphite crystals arranged axially along the fiber. The basic structural unit is a hexagonal network plane, and the atomic levels of carbon fibers are subject to irregular translation and rotation, so carbon fibers belong to the turbostratic graphite structure [88,89,90]. Inheriting the anisotropy of graphite lamellar structure, some physical properties of carbon fibers also show significant differences in axial and radial directions, especially the mechanical properties. According to length, carbon fiber may be separated into long, short, and short-cut fibers, all of which have superior corrosion resistance, thermal conductivity, coefficient of thermal expansion, axial strength, and modulus. As mentioned above, the addition of thermal conducting particles often sacrifices the mechanical properties of composites. To solve this problem, the researchers took advantage of the excellent mechanical properties of short-cut fibers in the axial direction to achieve simultaneous improvement in thermal conductivity and mechanical properties. In the process of FFF 3D printing, shear stress will occur between the melt and the inner wall of the nozzle. In order to reduce the flow resistance, the carbon fibers are forced to arrange along the flow direction, thus forming a phenomenon of orientation along the printing direction. Due to the shape and orientation of carbon fibers, when the material is stressed, the crack growth direction is perpendicular to the fiber orientation, which limits the crack growth while converting part of the stress, thus improving the mechanical properties of the composite. As shown in Figure 12, Liao et al. [70] prepared a CF/polyamide (PA) 12 composite material that could be used in FFF. The material’s tensile and flexural strength improved by 102.2% and 251.1%, respectively, when the CF filling quantity was 10 wt%. At the same time, the thermal conductivity of the vertical sample (type B) was improved by 277% due to the continuous heat conduction channel established by the presence of carbon fiber. Due to the unique orientation of carbon fiber, the impact strength of the composite exceeds that of pure PA12 when the load reaches a certain level, which also solves the problem of the weak impact strength of carbon fiber. Many studies have shown similar trends, but there are still some problems with carbon fiber thermally conductive materials prepared using FFF. For example, when the carbon fiber content is too high, the decreased fluidity of the melt can lead to blockage of the nozzle, which will limit the improvement of the material’s thermal conductivity. 

### 4.2. Insulating Thermally Conductive Fillers

Research on polymer-based materials with excellent thermal conductivity and insulating properties is a hot topic for further development in electrical and electronic fields. Insulating thermally conductive materials can protect electronic components on the one hand and export the heat generated by integrated circuits in time to ensure the safe operation of electronic devices on the other. Insulation and thermal conductivity fillers mainly include metal oxides (Al_2_O_3_ [91], SiO_2_ [92], etc.) and metal nitrides (AIN [93], BN [94], etc.).

#### 4.2.1. BN

In the crystalline structure of BN, B and N atoms form a densely connected hexagonal ring network within each layer due to strong covalent bonds and dipole moment forces, whereas the layers are connected by van der Waals forces and electrostatics [95]. The crystal structure can be divided into hexagonal and cubic crystal types by stacking heterogeneous atoms between layers [96,97]. Under high temperature and high pressure, hexagonal crystal form can be transformed into cubic crystal form, including four variants: hexagonal boron nitride (hBN), rhombic boron nitride (RBN), cubic boron nitride (CBN), and wurtzite boron nitride (WBN). Among them, hBN is referred to as “white graphite” since it is white and has a similar lamellar structure to graphene [98]. Similar to graphene, hBN has a high thermal conductivity of 600 W∙m^−^^1^∙K^−^^1^ in the in-plane direction, yet only 30 W∙m^−^^1^∙K^−^^1^ in the through-plane direction [99]. Due to its low dielectric constant, low dielectric loss, and high volume resistance, hBN is not only an excellent conductor of heat but also an excellent insulator of electricity. It is very suitable for electronic packaging materials, which can not only provide mechanical support for electronic chips but also achieve the purpose of heat dissipation.

Studies have shown that the thermal conductivity of hBN/polymer composites is influenced by various factors, including the orientation of the hBN, the quantity of filling, and the interfacial bonding between the hBN and the matrix. To increase the thermal conductivity of TPU composites, Liu et al. [75] used the shear force generated by the nozzle during the FFF process to promote the alignment of hBN along the printing direction. As shown in Figure 13, the thermal conductivity of the samples printed by FFF has improved significantly, showing stable thermal conductivity even at 100 °C. When the filling amount was 40 wt%, the thermal conductivity of the printed direction sample (PD) was 2.56 W∙m^−1^∙K^−1^, compared with only 0.24 W∙m^−1^∙K^−1^ for the hot-pressed sample of pure TPU, an improvement of 966.66%. At the same time, the coefficient of thermal expansion of hBN in the in-plane direction is negative at room temperature, which can offset the expansion phenomenon of TPU during heating and solve the deformation and warping problems of printed samples. The thermal conductivity of the printed sample prepared by Tyler et al. [76] by adding 35 wt% BN was only 0.93 W∙m^−1^∙K^−1^, which was improved but still failed to meet the requirements of thermal conductivity materials in engineering applications (>1 W∙m^−1^∙K^−1^). In this regard, the authors propose that this is due to the poor adhesion of BN to the ABS substrate, and that the thermal conductivity can be further improved by the subsequent surface modification of BN. 

#### 4.2.2. SiC

Si atoms and C atoms are covalently bonded by SP^3^ to form a tetrahedron, which constitutes the basic constituent unit of the SiC crystal. SiC has the advantages of broad band gaps and high electronic saturation rates, so its development prospects in the semiconductor industry are promising. In addition, SiC has high thermal stability and corrosion resistance [100], and its theoretical thermal conductivity can reach 490 W∙m^−1^∙K^−1^ [101]. As the development of the electronics industry has progressed, SiC has solved the problem of heat dissipation under high heat densities, thereby prolonging the device’s service life. Liu et al. [80] used SiC and C as the fillers of PLA to make the shape memory polymer, placed the composite material in hot water to trigger the shape recovery, and evaluated the shape recovery properties of materials with different filler contents by recovery speed and recovery time. With the addition of thermal conductivity filler, the maximum deformation recovery time of the composites is significantly reduced from 1.9 s to 0.25 s with the addition of 50 wt% C and 10 wt% SiC, which is mainly attributed to the high thermal conductivity of 4.777 W∙m^−1^∙K^−1^. This work shortens the response time of shape-memory polymers by regulating the thermal conductivity of the materials, which can be used to design the structures of shape-memory materials activated at different rates.

#### 4.2.3. Diamond

The fundamental structural particles of diamonds are carbon atoms, where each carbon atom is linked to four carbon atoms in sp3 hybrid orbitals to form a tetrahedron. Each carbon atom is located at the center of the tetrahedron, and the surrounding four carbon atoms are at the vertices. Because of the high bond energy of the C-C bond makes diamond the most rigid solid in nature [102]. Since the valence electrons of carbon atoms are used to form covalent bonds, resulting in no free electrons in diamond, its volume resistivity is as high as 5 × 10^14^ Ω cm, which is an insulating material. In terms of heat conduction, diamond mainly transfers heat through lattice vibration, and phonon scattering is small, so the thermal conductivity is as high as 2000 W∙m^−1^∙K^−1^ [103,104], and the coefficient of thermal expansion is only (0.86 ± 0.1) × 10^−5^/K, which makes it an excellent choice of thermal conductivity material. In addition, diamond has excellent optical and mechanical properties and chemical corrosion resistance, so it has great potential for heat dissipation.

As shown in Figure 14, Waheed et al. [105] dissolved the diamond and ABS in acetone and underwent six extrusions, ensuring the diamond’s uniformity and making the wire more dimensionally stable. After several times of mixing, the upper limit of the thermally conductive filler quantity is increased, thereby considerably enhancing the heat dissipation performance of the print. The disadvantage is that even with a high content of excellent thermal conductivity fillers, the greatest thermal conductivity of the composite is only 0.94 W∙m^−1^∙K^−1^, leaving a great deal of space for improvement. In this work, diamond and ABS are only physically combined and not chemically bonded, which may be a significant factor in the composites’ low thermal conductivity. Su et al. [106] utilized octadecyl amine (ODA) as the surfactant of diamond to form a hydrogen bond between the -NH_2_ group of ODA and the -OH group of diamond, thereby enhancing the interface compatibility between filler and matrix and promoting phonon transfer between diamond and PLA. The maximum thermal conductivity attained in this work is 2.22 W∙m^−1^∙K^−1^, which can guide the future research of FFF thermally conductive materials.

## 5. FFF Thermally Conductive Composites Process Parameters

For conventional FFF printing materials, extensive theoretical research and practical experience have produced a relatively comprehensive data model. However, to maximize the composite’s thermal conductivity after the addition of thermally conductive fillers, optimization of parameters in each process is essential. In every work, parameter optimization is often used to prepare for subsequent work, as the change in parameters directly determines the internal structure of the printed part and thus affects the thermal conductivity. The most common printing parameters and their intrinsic relationship to thermal conductivity are listed below.

### 5.1. Nozzle Temperature

As the nozzle is in direct contact with the polymer, the nozzle’s temperature will determine the fluid’s melt state [107]. The polymer is not entirely molten when the nozzle temperature is too low. When the temperature is increased appropriately, the flow of the polymer improves. At this point, the diffusion of interlayer polymer chains increases, and the filament extruded from the nozzle will transport heat to the deposited part, forcing it to remelt, ensuring better interlayer adhesion, and enhancing thermal conduction throughout the entire print [108]. When the temperature rises, it provides a longer time for the crystallization process of the polymer so that the crystallinity will increase. The orderliness of the molecular chains in the crystalline region increases, the resistance decreases when phonons diffuse, and the propagation velocity from one end to the other becomes faster, manifested as an increase in thermal conductivity. However, high nozzle temperatures may lead to polymer degradation or deformation of the filament during deposition due to the inability to reduce the temperature quickly, making the printed part less accurate. Therefore, the proper nozzle temperature should ensure the complete melting of the filament and maintain the extruded filament without deformation.

### 5.2. Nozzle Diameter

The diameter of the nozzle has an effect on the shear force in the nozzle, with the following formula [75]:$$\tau =\frac{32{\eta \rho Q}_{v}}{{\pi d}^{3}}$$
where η is the viscosity of the melt, ρ is the density of the filament, d is the diameter of the nozzle, and Q_v_ is the flow rate. η, ρ, and d can be regarded as constants for the same material under the same experimental conditions. As shown in the formula, nozzle diameter and shear force are inversely related, and as the diameter rises, the shear force falls. As previously noted, the shear force produced during the printing process causes anisotropic fillers to align along the printing direction, resulting in the formation of a thermally conductive network. As a result, the filler’s degree of orientation is determined by the shear force, which impacts thermal conductivity.

### 5.3. The Printing Speed

Print speed affects the mechanical properties and accuracy in the printed part, and in the case of thermal conductivity, improper print speed can create more voids that affect thermal conductivity. When the printing speed is too slow, on the one hand, it makes printing inefficient, and on the other hand, the extruded melt will be stacked together, making the accuracy decrease. If the printing speed is too fast, the filament will be sent out before it is completely melted, and the extrusion speed of the melt cannot keep up with the printing speed, which will cause an uneven diameter of the filament, resulting in voids inside the parts. Therefore, a suitable printing speed should ensure a specific printing efficiency and the preparation of printed parts with a dense internal structure.

### 5.4. Platform Temperature

The platform temperature refers to the temperature of the print filament deposition plane, which directly affects the mechanical properties and molding quality of the printed part. If the platform temperature is too low, the extruded filaments will not deposit appropriately on the platform, resulting in warping and delamination of the print. After the temperature of the platform is moderately increased, the bottom layer will conduct the temperature to the middle layer. After a while, the temperature of the middle layer decreases to a more stable temperature. After incorporating the thermally conductive filler, the heat transfer between the layers is enhanced, thus increasing the stable temperature of the middle layer. If the temperature of the interlayer can be kept above Tg for a longer time, the diffusion of the polymer chains between the layers can be increased, thus improving the adhesion of the layers [65]. Simultaneously, the cooling time of the print following an increase in platform temperature is lengthened, which facilitates the remelting and recrystallization of the deposited filaments, thereby strengthening interlayer bonding and decreasing heat conduction resistance. However, the temperature of the platform must not be too high, or the natural cooling and shaping of the print would be affected.

## 6. Ways to Improve Thermal Conductivity

In order to improve the thermal conductivity of FFF 3D-printed parts, we can start with other factors in addition to changing the printing parameters. Table 3 summarizes the methods to improve the thermal conductivity of FFF 3D prints, followed by a brief introduction of related studies. 

### 6.1. Porosity

Given that the thermal conductivity of air at room temperature is only 0.0242 W∙m^−1^∙K^−1^, the existence of voids will affect the thermal conductivity of the print. Many studies have focused on reducing porosity by adjusting parameters and designing printing models. In order to create prints with low porosity, Guo et al. [68] adopted a multiscale dense structure design. First, they used graphene made by physical exfoliation, so the filler was denser and less porous. The accuracy of the print was then increased by optimizing the platform temperature and printing speed. Finally, the extrusion multiplier was increased to 1.2, which enhanced the interfacial adhesion between the printed filaments, resulting in solid connections between adjacent filaments. Jia et al. [63] did not take the approach of reducing the porosity but designed the printed model to achieve maximum heat transfer. As shown in Figure 15, the deposition angle of the filaments in stand-3D printing (SP) and flat-3D printing (FP) is different, where a large number of voids in FP blocks the transfer of heat across the plane, so the thermal conductivity does not meet industrial requirements even at 50 wt% graphene filling (>1 W∙m^−1^∙K^−1^). In contrast, the heat conduction channels formed by the pores in the SP are aligned with the direction of heat transfer, and the thermal conductivity of the SP is more than five times higher than that of the FP at the same load. This method allows heat to pass smoothly through the channels between the deposited filaments, reducing the heat loss from the pores. Thus, in order to reduce the effect of pores on thermal conductivity, it is possible to prepare tightly structured prints by reducing the porosity, and also to design the orientation of the pore channels so that heat can be transported smoothly.

### 6.2. Dispersion

Increasing the dispersion with a constant filler content can reduce the agglomeration among the fillers and maximize the thermal conductivity network or channels. In addition, it can improve the interfacial compatibility and adhesion between filler and matrix and solve the extrusion problem of composite materials under high loading. As demonstrated in Figure 16, Jing et al. [66] utilized the strong shear field of solid-state shear milling to exfoliate GNPs into many layers while simultaneously improving the dispersion of GNPs. Due to the planar structure of graphene sheets, the increased dispersion enhances the interfacial coupling between GNPs and LLDPE, resulting in a significant reduction in the thermal boundary resistance. As the percentage of GNPs increases, the thermal conductivity advantage of samples treated with solid-state shear milling becomes increasingly evident. In addition to mechanically mixing the matrix and filler before melt mixing, researchers have utilized various mixing techniques to increase the dispersion of thermally conductive particles. Rostom et al. [65] combined solution mixing and melt mixing, dissolved PLA and graphene in DMF, and utilized the principle of PLA’s low solubility in water to rapidly precipitate PLA and trap graphene sheets in order to maintain a good dispersion of graphene sheets. Guo et al. [68] also performed the same pretreatment of solution mixing, with the difference being that they directly dried the dissolved viscous solution, cut it into small pieces, and fed it into a twin-screw extruder for extrusion to further achieve filler dispersion by double mixing.

### 6.3. Filler Content

In FFF 3D printing, most studies have shown that thermal conductivity positively correlates with filler content. When the filler content is too small, the thermally conductive particles are encapsulated by the polymer matrix in an island-like form, and the filler and the polymer can be seen as two thermal resistances in series. When there are more thermal conducting particles, the thermal conducting network penetrates each other. The polymer and fillers are equivalent to parallel connection in the heat flow direction, and the thermal resistance decreases significantly. Jing et al. [66] investigated the distribution and connectivity of thermally conductive fillers in printed parts with three graphene contents. When the volume fraction is less than 7 vol%, graphene is only distributed in LLDPE alone, so the thermal conductivity enhancement of the composite is not significant. The individually distributed graphene sheets will come into contact with one another in some spots to form short-range conductive bridges when the volume percentage reaches 10%, which also results in a significant boost in thermal conductivity. The short-range aligned conductive bridges were interconnected to create GNP networks aligned in a wide range when the content was raised to 15 vol%. This structure can significantly reduce the thermal resistance at the interface between GNP and LLDPE, which helps enhance prints’ thermal conductivity. However, the increase in filler amount will raise the cost and bring a series of problems, such as mechanical performance degradation and nozzle clogging. Therefore, a practical working design should not blindly increase the filling amount but achieve optimal performance with less cost.

### 6.4. Filler Orientation

Due to shear force, anisotropic fillers, such as graphene, graphite flakes, BN, and carbon fibers, will be orientated when passing through the nozzle. A higher degree of orientation contributes to forming an efficient thermal conduction path and reducing the thermal interface resistance between the filler and the matrix to maximize the thermal conductivity at a lower filling quantity. Due to the anisotropy of thermal conductivity, the orientation direction typically has a higher thermal conductivity than the non-orientation direction. If this point can be used to design the orientation of the filler such that heat in the heat dissipation material may be rapidly transmitted via a highly oriented heat conduction network, the heat dissipation performance of the material will be significantly enhanced. In order to raise the degree of anisotropic fillers’ orientation, the researchers utilized various methods to boost the shear force of the printing process.

Zhu et al. [64] discovered that GNPs tended to be aligned along the printing direction in printed samples with a raster angle of 90°, and the degree of orientation of GNPs was calculated to be approximately 76.4° based on theoretical calculations. In contrast, the GNP in the compression molded (CM) samples is disordered, creating a stark contrast. However, they did not conduct further studies on the relationship between nanofiller orientation and printing parameters. Liu et al. [75] prepared two different samples along the printing and thickness directions. In the PD sample, the hBN sheets were arranged more along the vertical direction, and the thermal conductivity of the PD sample was higher than that of the TD sample because the hBN was parallel to the heat flux direction. The degree of hBN orientation was deduced from testing the variation in thermal conductivity under various nozzle sizes and printing speeds. As the nozzle diameter rises, the material’s degree of orientation reduces. In this work, the printing speed has little effect on the thermal conductivity.

## 7. Summary and Outlook

This paper summarizes the fundamental principles of several conventional 3D printing technologies and their research advancements in heat conduction. Since the 1990s, FFF 3D printing technology has increasingly matured and been applied in numerous fields, such as aerospace, medical treatment, education, etc.

In addition, two mechanisms of thermal conductivity of polymers are briefly described in this paper. However, the polymer’s limited thermal conductivity severely restricts its application. To improve the thermal conductivity of polymer matrix composites, thermally conductive fillers are often added. These materials are manufactured using FFF 3D printing technology that offers more significant advantages in terms of equipment price and ease of operation. There have been many studies on improving the thermal conductivity of materials based on FFF 3D printing, which can be roughly divided into two ways. The first method is to directly mix the high-thermal-conductivity filler with the polymer and minimize the heat conduction resistance by adjusting the content and orientation of the filler to form the thermal conduction channel in the matrix. This method is usually only physical mixing, and there is no chemical bond between the filler and the matrix, so the improvement of thermal conductivity is limited. The second is to further disperse the filler before melt mixings, such as predispersion by liquid mixing, solid-state shear milling, or separation network construction by specific chemical reactions. This approach is advantageous in increasing the compatibility between the filler and the matrix, decreasing the thermal resistance at the interface, and improving thermal conductivity more obviously than physical mixing.

Currently, polymer-based thermally conductive materials prepared based on FFF 3D printing can be applied in the following fields. For new materials prepared by non-insulating thermally conductive fillers, customized 3D printing radiators can be designed by taking advantage of the shape designability of FFF, and the heat dissipation efficiency can be maximized through the design of radiator shape and structure. Thermally conductive materials made from insulating fillers can be used as microelectronic packaging materials, battery thermal management materials, and thermal interface materials to solve thermal management problems in advanced electronic devices. In meeting electronic equipment’s cooling requirements while achieving mechanical support, circuit protection, and insulation, it is expected to reach industrial application in ultrahigh-power electronic equipment. 

Although FFF thermally conductive composites offer considerable application potential, there are still some challenges to be resolved in this area of research:

(1) Due to the layer-by-layer nature of FFF printing, holes between filaments and layers are unavoidable and impede the smooth transfer of heat.

(2) There is a maximum filling quantity for thermally conducting particles. When the filling amount is too high, the mechanical qualities of the prints are greatly diminished. In contrast, when the filling amount is too low, the thermal conductivity of the prints is not significantly enhanced.

(3) The lowering of interface heat resistance remains a formidable challenge. Interfacial thermal resistance consumes a portion of the heat due to the inconsistent phonon frequencies between fillers and fillers or between fillers and matrix.

Due to the difficulties mentioned above, in most of the current work, the improvement of thermal conductivity of printed parts is still limited despite incorporating a large number of highly thermally conductive fillers. The research work on FFF thermally conductive materials requires the improvement of the compatibility of thermally conductive particles with the matrix at the material level and the minimization of pores that impede heat transfer through structural design. Future research needs to explore how to maximize the thermal conductivity with minimal filler content to achieve both cost reduction and reduce the sacrifice of the mechanical properties of the composite. In the current work, filler addition in the matrix tends to present an undifferentiated form. Future research can take full advantage of the layer-by-layer stacking nature of 3D printing to design thermal conductivity gradients or thermal conductivity pathways to address the problem of material damage due to thermal stresses generated by uneven heat at different locations.

## Figures and Tables

**Figure 1 polymers-14-04297-f001:**
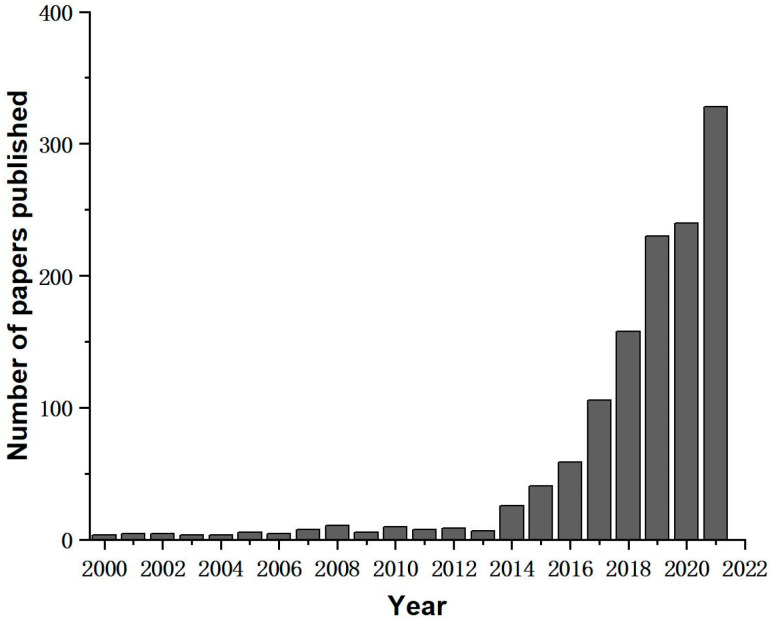
Publication of articles using 3D printing for thermal conductivity studies from 2000 to 2021. Source: Web of Science.

**Figure 2 polymers-14-04297-f002:**
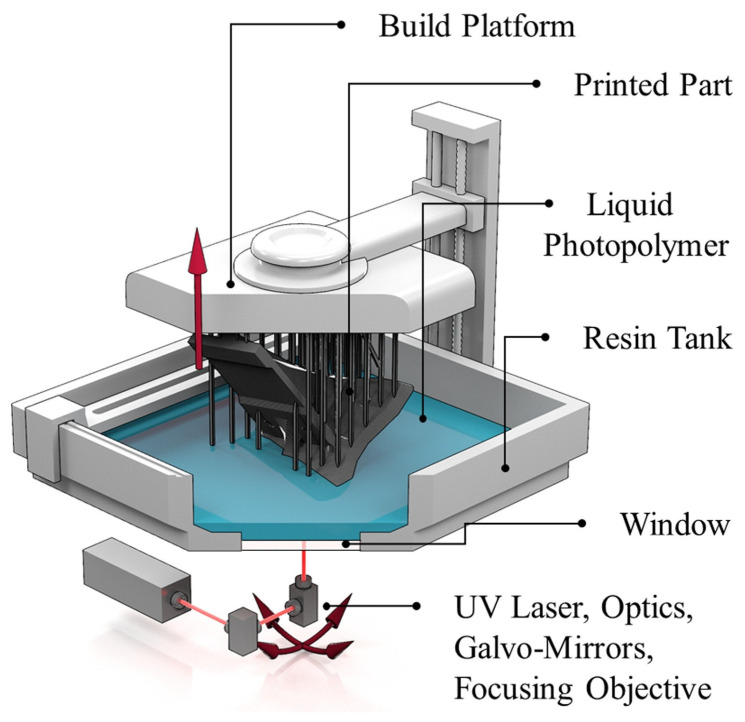
Schematic of 3D printing with SLA (reproduced with permission from Reference [21], Copyright 2020 Elsevier).

**Figure 3 polymers-14-04297-f003:**
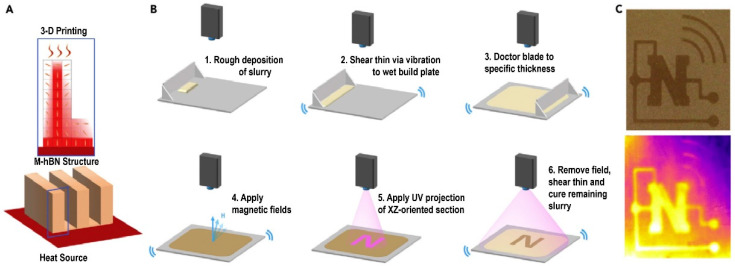
(**A**) SLA, magnetic field induction, and vibration are combined in a novel 3D printing process to create composites with specific thermal conduction pathways; (**B**) schematic of the fabrication of specific thermal pathways through a customized 3D printing procedure employing mhBN-acrylate as the raw material; (**C**) visible light (top) and thermal capture (bottom) of the composite, with mhBN oriented out-of-plane in the “N”-shaped and semiarc ring sections, which correspond to higher temperatures under the infrared camera (yellow) (reproduced with permission from Reference [34], Copyright 2020 Elsevier).

**Figure 4 polymers-14-04297-f004:**
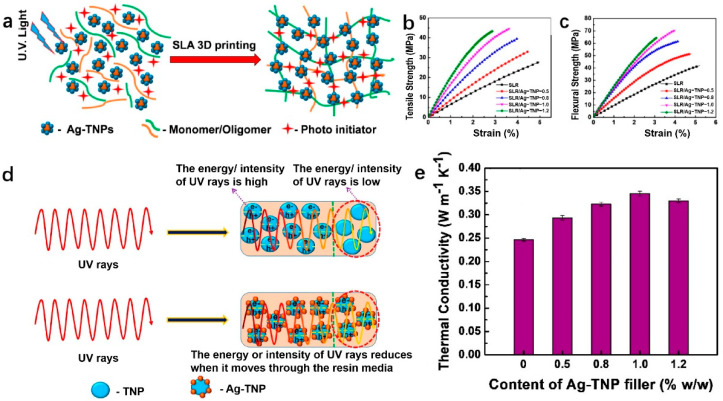
(**a**) Schematic illustration of SLR (stereolithographic resin) and Ag−TNP (Ag−ornamented TiO_2_ nanoparticles) under UV light via SLA−induced photopolymerization for 3D printing; (**b**) tensile strength and (**c**) tensile strength of pure SLR and different concentrations of SLR/Ag−TNP; (**d**) schematic diagram of the mechanism of SLR/Ag−TNP−induced photopolymerization; (**e**) thermal conductivity of pure SLR and different concentrations of SLR/Ag−TNP (reproduced with permission from Reference [31], Copyright 2020, Mubarak et al.).

**Figure 5 polymers-14-04297-f005:**
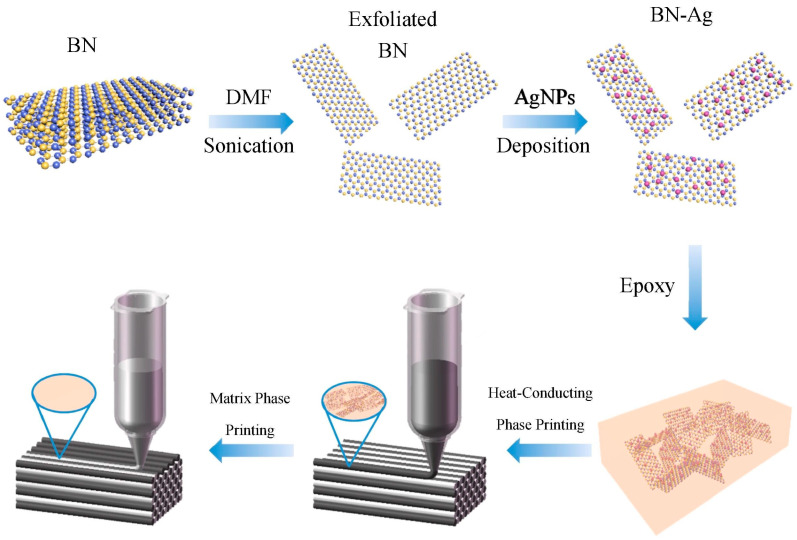
Schematic diagram of the preparation of BN-Ag/epoxy resin composites (reproduced with permission from Reference [37], Copyright 2020 Elsevier).

**Figure 6 polymers-14-04297-f006:**
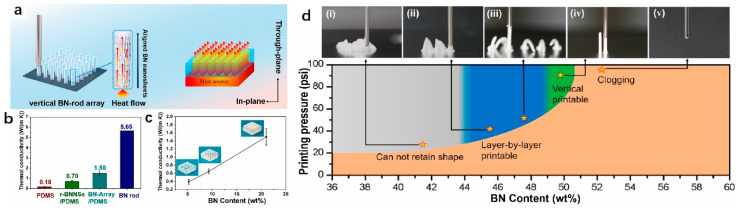
(**a**) The left image is a schematic illustration of the fabrication of BN arrays using DIW, the middle inset is a magnified aligned BN nanosheet, and the right image is a schematic illustration of thermal conduction along the vertically aligned BN sheets; (**b**) through-plane thermal conductivity of pure PDMS, random BN/PDMS, BN arrays/PDMS, and BN rods; (**c**) through-plane thermal conductivity of different numbers of BN rods/PDMS; (**d**) printability of extruded ink filaments at different BN contents, insets (**i**–**v**) showing the shape retention ability of vertical printing (reproduced with permission from Reference [38], Copyright 2019 American Chemical Society).

**Figure 7 polymers-14-04297-f007:**
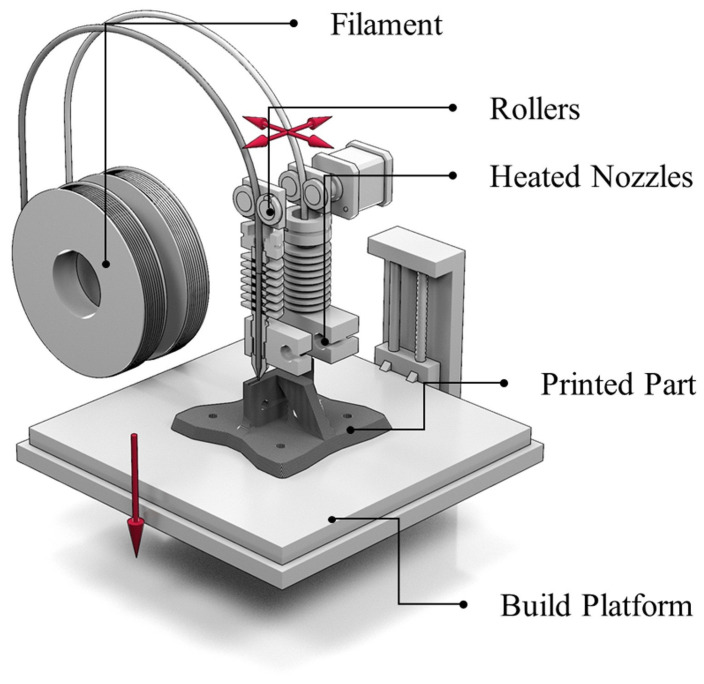
Schematic of 3D printing with FFF (reproduced with permission from Reference [21], Copyright 2020 Elsevier).

**Figure 8 polymers-14-04297-f008:**
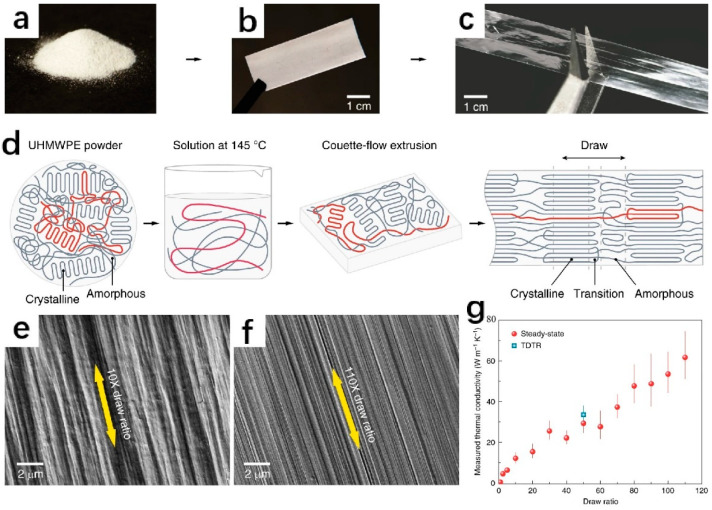
Optical photographs. (**a**) UHMWPE powder; (**b**) opaque extruded film; (**c**) transparent drawn film; (**d**) schematic diagram of the preparation process of a high-thermal-conductivity polymer film. SEM images of films with different draw ratios (**e**) with a draw ratio of 10; (**f**) with a draw ratio of 110; (**g**) thermal conductivity of polymer films at different draw ratios. The red dots are the steady-state experimental test results, and the blue square is the average of 20 transient thermal conductivities (reproduced with permission from Reference [62], Copyright 2019, Xu et al.).

**Figure 9 polymers-14-04297-f009:**
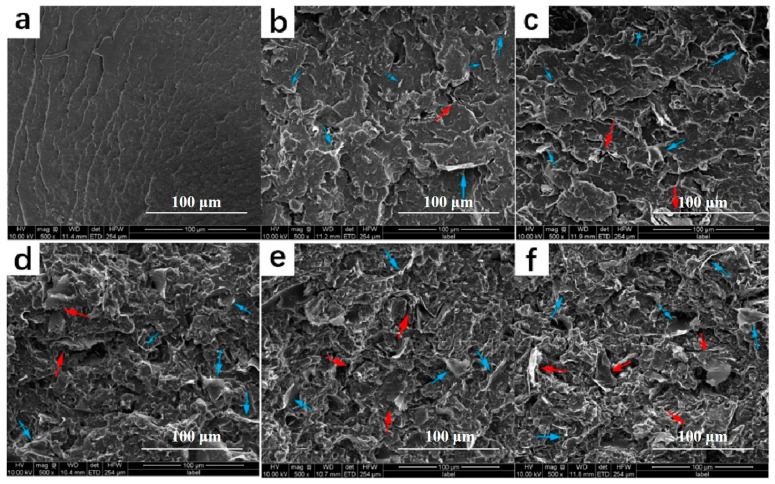
Scanning electron microscope image. (**a**) Pure PA12; (**b**) PA12/2 wt% GNPs; (**c**) PA12/4 wt% GNPs; (**d**) PA12/6 wt% GNPs; (**e**) PA12/8 wt% GNPs; (**f**) PA12/10 wt% GNPs (reproduced with permission from Reference [64], Copyright 2017 John Wiley and Sons).

**Figure 10 polymers-14-04297-f010:**
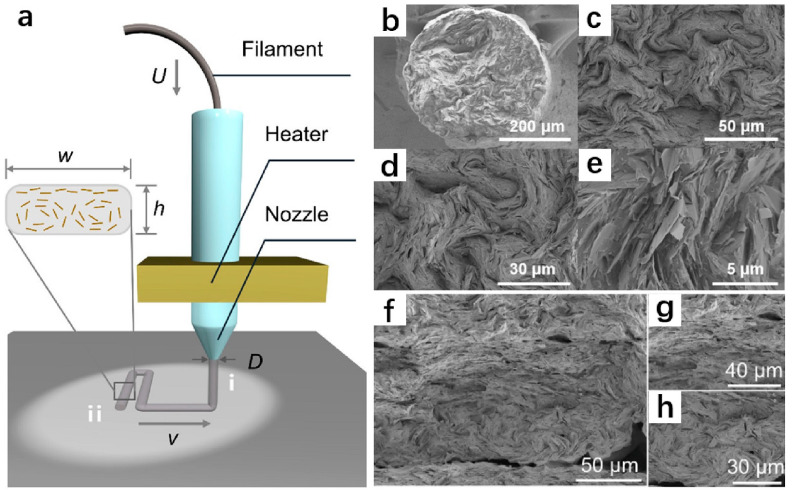
(**a**) Schematic illustration of the FFF printing process of graphene/TPU composites, including extrusion and deposition processes; SEM image of (**b**) a cross section of a 40% graphene/TPU filament; (**c**–**e**) enlarged view of the cross section of the filament; (**f**) asymmetric structure of the printed filament; (**g**) top-aligned graphene sheets; (**h**) graphene sheet with a helical bottom (reproduced with permission from Reference [68], Copyright 2021 American Chemical Society).

**Figure 11 polymers-14-04297-f011:**
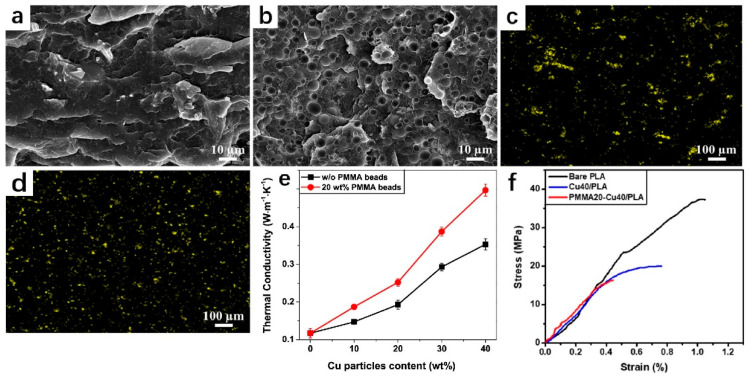
(**a**,**b**) SEM image of Cu40/PLA composite and Cu40/PLA/PMMA; (**c**,**d**) EDS mapping of Cu40/PLA composite and Cu40/PLA/PMMA; (**e**) thermal conductivity with different copper content in PLA/PMMA/Cu and PLA/Cu composites; (**f**) tensile curves of printed pure PLA and PLA composites (reproduced with permission from Reference [82], Copyright 2020 Elsevier).

**Figure 12 polymers-14-04297-f012:**
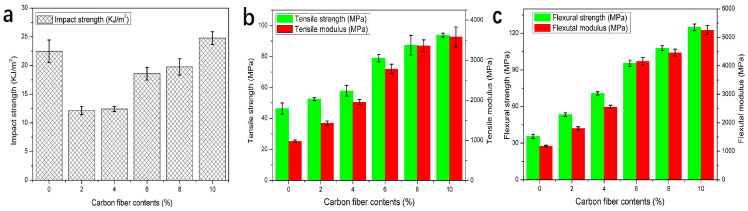
(**a**) Impact strength of PA12 and CF/PA12 composites; (**b**) tensile strength of PA12 and CF/PA12 composites; (**c**) flexural strength of PA12 and CF/PA12 composites (reproduced with permission from Reference [70], Copyright 2018 Elsevier).

**Figure 13 polymers-14-04297-f013:**
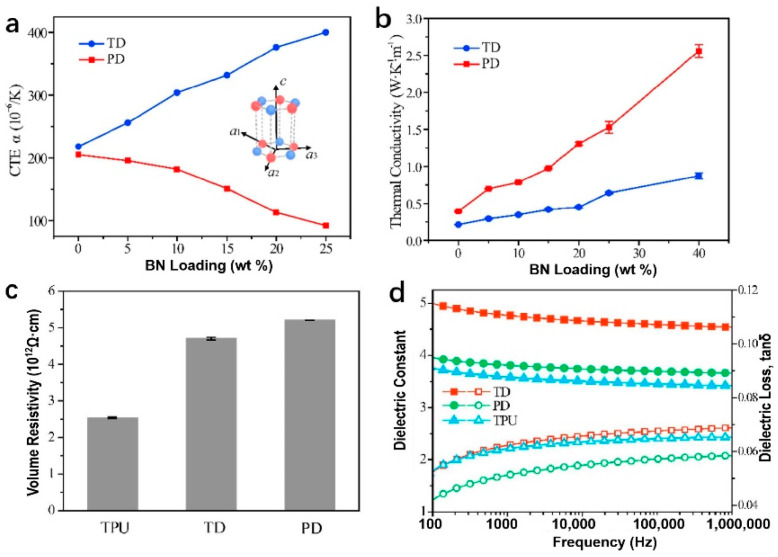
(**a**) The hBN/TPU composite’s coefficient of thermal expansion (CTE). The image embedded is a hexagonal cell; (**b**) thermal conductivity of hBN/TPU composites at 100 °C; pure TPU and 20 wt% hBN−loaded TPUs: (**c**) volume resistivity; (**d**) dielectric constant and dielectric loss (reproduced with permission from Reference [75], Copyright 2019 Elsevier).

**Figure 14 polymers-14-04297-f014:**
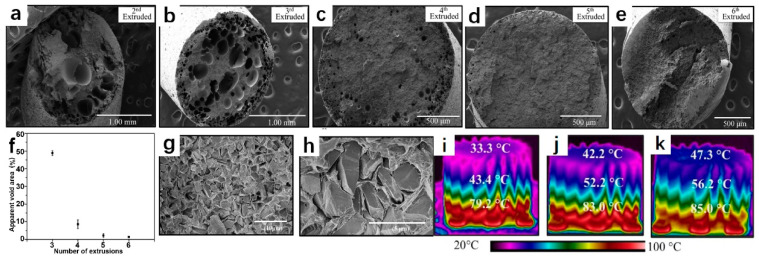
(**a**–**e**) SEM images of diamond/ABS filament cross section after 2–6 extrusions; (**f**) its apparent void area; (**g**,**h**) SEM images of diamond/ABS (60%) showing that the diamond is only physically connected to the polymer; (**i**–**k**) temperature distribution of 3D-printed heat sinks made of ABS, D-ABS (37.5%), D-ABS (60%) after heating for ten minutes (reproduced with permission from Reference [105], Copyright 2019 American Chemical Society).

**Figure 15 polymers-14-04297-f015:**
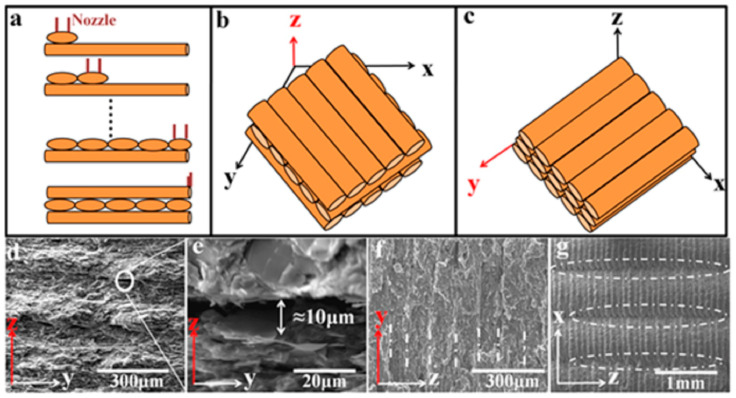
(**a**) Schematic illustration of void generation during FFF; (**b**,**c**) schematic illustration of void formation in FP and SP; (**d**–**f**) the voids of FP (**d**,**e**) and SP with the same filler content (**f**,**g**) (reproduced with permission from Reference [63], Copyright 2017 Elsevier).

**Figure 16 polymers-14-04297-f016:**
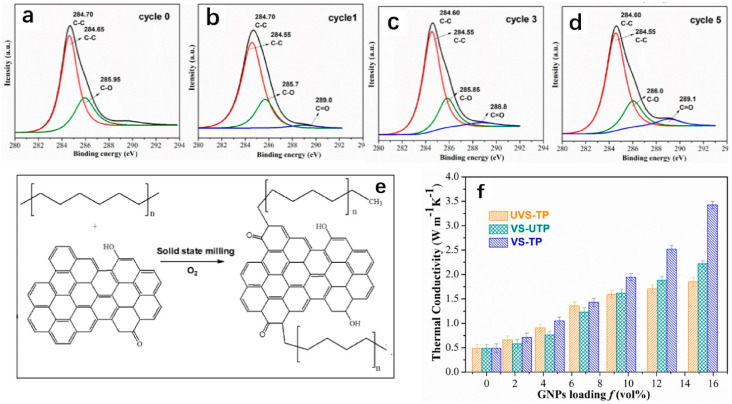
(**a**−**d**) XPS curves of LLDPE/GNPs after different milling times ((**a**): 0 milling; (**b**): 1 milling; (**c**): 3 millings; (**d**): 5 millings); (**e**) schematic diagram of mechanochemical reaction during solid−state shear milling; (**f**) thermal conductivity (TC) of FFF 3D prints at different GNP content (TPTC of vertically printed samples after solid−state shear milling (VS−TP), TPTC of vertically printed samples without solid−state shear milling (UVS−TP), and IPTC of vertically printed samples after solid−state shear milling (VS−UTP) (reproduced with permission from Reference [66], Copyright 2020 Elsevier).

**Table 1 polymers-14-04297-t001:** Summary of thermally conductive composites prepared by DIW.

Matrix	Filler and Content	TC (W∙m^−1^∙K^−1^)	Ref.
MEP (modified epoxy)	20 wt% BN	2.52	[37]
Pluronic F127	21 wt% BN	1.5	[38]
EPON Resin 862	7.6 wt% carbon fiber and 9.3 wt% graphite	~2	[39]
Poly(lactic-co-glycolic acid)	60 vol% hBN	2.1	[40]
Epoxy	3 wt% graphene oxide	~0.7	[41]
CPE (composite polymers electrolytes)	2 wt% silane-treated hBN (S-hBN)	1.031	[42]
Epon 826 Epoxy Resin	2.55 vol% BN	0.283	[43]
Nitrile Rubber Latex	39.1 wt% glass fibers and 0.39 wt% graphene	1.2	[44]

**Table 2 polymers-14-04297-t002:** Summary of FFF thermally conductive fillers.

Filler	Matrix	Loading	TC (W∙m^−1^∙K^−1^)	Other Improved Properties	Ref.
Graphite	PA6/POE-g-MAH/PS	50 wt%	5.5	------	[63]
Graphene nanoplatelet (GNPs)	PA12	10 wt%	1.12	Elastic modulus	[64]
Graphene	PLA	2 wt%	0.24	Modules and ultimate strength in the Z direction	[65]
GNPs	LLDPE	15.0 vol%	3.43	Tensile strength and Young’s modulus	[66]
Expandable graphite	PE	20 wt%	2.71	------	[67]
Graphene	TPU	20 wt%	12	------	[68]
GNPs	PLA	12 wt%	0.662	------	[69]
Carbon fiber	PA 12	10 wt%	0.835	Tensile strength and flexural strength	[70]
Carbon fiber	PLA	5 wt%	0.1342	Tensile strength and flexural strength	[71]
GNPs + MWCNTs	PLA	4.5 wt% + 1.5 wt%	0.4692	Electrical conductivity	[72]
Graphite nanoplates (GnP) + MWCNTs	PEEK	5 wt% + 3 wt%	0.5	Electrical conductivity	[73]
BN	PLA	50 wt%	2.463	------	[74]
AIN	PLA	70 wt%	1.543	------	[74]
BN	TPU	40 wt%	2.56	------	[75]
BN	ABS	35 wt%	0.93	------	[76]
BN	PA 6	30 vol%	2.03	------	[77]
BN	PCO	20 wt%	0.55	Electrical conductivity	[78]
BNNT	PS	10 wt%	0.382	Thermal stability	[79]
SiC + C	PLA	50 wt% + 10 wt%	4.78	Shape recovery ability	[80]
Cu	ABS	50 wt%	0.912	Tensile modulus	[81]
Cu	PLA + PMMA	40 wt%	0.49	Elastic modulus and thermal stability	[82]
Cu	ABS	30 vol%	~3.7 (220 °C)	Mechanical properties	[48]
Fe	ABS	45 vol%	~1.86	Mechanical properties	[48]

**Table 3 polymers-14-04297-t003:** Summary of methods to improve the thermal conductivity of FFF 3D prints.

Employed Methods	Purpose	Matrix	Filler	TC (W∙m^−1^∙K^−1^)	Ref.
Designing the printed model	Aligning pore direction with heat flow	PA6/POE-g-MAH/PS	Graphite	5.5	[63]
Changing raster angle	Increasing the degree of orientation	PA12	GNPs	1.12	[64]
Double mixing of solution and melt	Facilitating dispersion	PLA	Graphene	0.24	[65]
Solid-state shear milling and increasing content	Exfoliating GNPs and building thermal conductivity networks	LLDPE	GNPs	3.43	[66]
Designing multiscale dense structures	Reducing porosity	TPU	Graphene	12	[68]
Changing print direction and reducing nozzle diameter	Increasing the degree of orientation	TPU	BN	2.56	[75]

## Data Availability

Not applicable.

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
