# Peer review of "Progress of Polymer-Based Thermally Conductive Materials by Fused Filament Fabrication: A Comprehensive Review"

_polymers, 2022, doi:10.3390/polym14204297_

Round 1
Reviewer 1 Report
Reviewer comment :
1. Introduction
-The aim of the paper is to provide an understanding of the recent progress of different 3D printing technologies in heat conduction and the thermal conduction mechanism of polymer matrix composites. However, in the introduction, the purposes were stated in the latest paragraph. There is an absence of flow for the transfer of ideas;
- What are the gaps of polymer-based thermally conductive materials prepared by FFF before coming to the aims?
- Why this investigation is important and would be contribute to the domain?
- How does it differentiate from the other reviews?
The introduction needs to be revised to consider these points.
2. 3D Printing in Thermally Conductive Composites
Page 3, second paragraph of section 2.1, line 103; the author said “ Due to advantages of high precision and rapid prototyping, SLA is regarded as one of the manufacturing technologies of thermal management materials by researchers”.
Any information about the materials utilized for the SLA process. What polymers are used? In the SLA process, which polymers are more compatible with ceramic and metallic fillers? It will be better to add this information.
3. Thermal Conduction Mechanism
Page 9, first paragraph, line 263; the author said “Currently, thermal conduction network theory and percolation thermal conduction theory are the most well-known in the research field”.
Which polymers correspond to the thermal conduction network mechanism and what about the percolation thermal conduction mechanism?
It is more relevant to present comparatively way the advantages and disadvantages of each mechanism regarding the employed materials and manufacturing process.
6. Ways to improve thermal conductivity
There are many factors mentioned in this section. These issues could be synthesized (employed process, utilized polymer/copolymer, filler, outputs) and summarized in table form.
Author Response
Response to Reviewer 1 Comments
Dear Reviewer:
Thank you for your valuable comments on our paper entitled " Progress of Polymer-Based Thermal Conductive Materials by Fused Filament Fabrication: A Comprehensive Review" (Manuscript ID: polymers-1941164). Those comments are all valuable and very helpful for revising and improving our paper. We have studied comments carefully and have made correction which we hope meet with approval. The corrections are marked in red in the manuscript. The major revisions in the paper and responses to your comments are listed below.
Point 1:
- Introduction
-The aim of the paper is to provide an understanding of the recent progress of different 3D printing technologies in heat conduction and the thermal conduction mechanism of polymer matrix composites. However, in the introduction, the purposes were stated in the latest paragraph. There is an absence of flow for the transfer of ideas;
-What are the gaps of polymer-based thermally conductive materials prepared by FFF before coming to the aims?
-Why this investigation is important and would be contribute to the domain?
-How does it differentiate from the other reviews?
The introduction needs to be revised to consider these points.
Response 1:
Thank you for your suggestions on our manuscript. Your comments were very useful to us. Although the preparation of thermally conductive materials using FFF has the advantages of low cost and easy induction of filler orientation, the following problems remain in the current research. 1) There are inevitable voids in FFF prints that will hinder heat transfer. (2) The interfacial thermal resistance between the filler and the polymer matrix affects the final thermal conductivity of the product. (3) At present, the higher thermal conductivity of the prints is added to a lot of fillers, which increases the production cost on the one hand, and the mechanical properties of the prints will be decreased on the other hand. The above problems still need further research, so I have reviewed the research on the preparation of polymer-based thermally conductive materials by FFF, hoping to provide ideas from it to solve the above problems. As electronic devices continue to upgrade, the power is also increasing, so heat dissipation has become an unavoidable problem. It will affect the life of electronic products and even endanger users' lives. Therefore, taking advantage of FFF's short production cycle and customizable shape to prepare materials with high thermal conductivity has great significance. The products can be used in small electronic products and high-power devices. Unlike other reviews, this paper reviews the research progress of FFF 3D printing materials based on the conductive properties of thermally conductive fillers and summarizes the application directions of the materials based on this classification at the end of the paper. The internal relationship between the FFF process parameters and the thermal conductivity of the composites was also analyzed in detail. Summarized the general methods to improve the thermal conductivity of FFF materials and provided a reference for future research.
Therefore, I have made revisions on page 2, lines 66-81, 82, and 85-89.
Point 2:
- 3D Printing in Thermally Conductive Composites
Page 3, second paragraph of section 2.1, line 103; the author said “ Due to advantages of high precision and rapid prototyping, SLA is regarded as one of the manufacturing technologies of thermal management materials by researchers”.
Any information about the materials utilized for the SLA process. What polymers are used? In the SLA process, which polymers are more compatible with ceramic and metallic fillers? It will be better to add this information.
Response 2:
We greatly appreciate the reviewers' suggestions for the manuscript. This section lacks a description of the raw materials used in the SLA process, and to address this issue, and we outline the main components of SLA printing raw materials and their functions. Oligomers play a major role in the performance of the final printed part as the main component of the printing material and list the commonly used oligomers. The selection conditions for oligomers are relatively demanding, resulting in a relatively small selection of polymers available. After curing, the resin has poor toughness and mechanical properties and adding fillers to modify the resin is a solution. Since there are fewer choices of polymers, researchers often modify nanofillers to improve the compatibility of the filler with the substrate and the performance of the prints after curing. With this in thoughts, we added the following on page 4, lines 118-133: "The printing material of SLA, also known as photosensitive resin, usually consists of oligomers, reactive diluents, photoinitiators, etc. Oligomers are low molecular weight monomers or prepolymers that serve as the photosensitive resin's main components and determine the printed part's performance after curing. Choosing oligomers requires consideration of the physical and chemical properties of the cured product on the one hand and the feasibility of production, such as system viscosity and cost, on the other hand, so there are relatively few options. The most used oligomer in SLA is epoxy acrylate, followed by urethane acrylate. Other oligomers, such as polyester acrylate and polyether acrylate, can also be used as raw materials for photosensitive resins. Active diluents reduce the viscosity of oligomers and accelerate the reaction of resins. The photoinitiator generates reactive intermediates by absorbing radiation energy to activate oligomers and diluent monomers for cross-linking reactions. Since our research on SLA technology is late, the research on photosensitive resin is mainly concentrated in universities and research institutes. As the prints made by SLA have poor mechanical properties and toughness, the researchers used nanoparticles to modify the resin. It can improve the compatibility between the polymer and the filler and further enhance the prints' performance.”
Point 3:
- Thermal Conduction Mechanism
Page 9, first paragraph, line 263; the author said “Currently, thermal conduction network theory and percolation thermal conduction theory are the most well-known in the research field”.
Which polymers correspond to the thermal conduction network mechanism and what about the percolation thermal conduction mechanism?
It is more relevant to present comparatively way the advantages and disadvantages of each mechanism regarding the employed materials and manufacturing process.
Response 3:
We thank the reviewers for their comments on the content of the thermal conduction mechanism. In FFF 3D printing, highly thermally conductive fillers are usually added to improve the thermal conductivity of polymers. To better explain the thermal conductivity behavior of filled polymers, researchers have proposed the thermal conduction network theory and the percolation thermal conduction theory. These two theories are not specific to a particular polymer but have more relevance to the filler. The thermal conductivity network theory suggests that the thermal conductivity of a material is mainly related to the content of fillers. Therefore, if we want to improve the material's thermal conductivity, we can increase the content of filler, which is a relatively simple method. However, the disadvantages are also obvious. Increasing the filler content will increase the production cost and decrease the mechanical properties of the prints.
Percolation thermal conduction theory is mainly used to explain the thermal conductivity behavior of composites with the addition of ultra-high thermal conductivity fillers. The thermal conductivity increases substantially when the filler content is near the permeability value. When the filler content continues to increase, most materials' thermal conductivity increases linearly. This law can guide production and reduce costs while achieving the target thermal conductivity. Therefore I made revisions on page 10, lines 292-299, lines 309-311, lines 313-316, and lines 322-329, respectively. It is added that the thermal conduction network theory is suitable for explaining composites with added anisotropic fillers and the percolation thermal conduction theory is suitable for explaining composites with added ultra-high thermal conductivity fillers. The advantages and disadvantages of these two theories are also discussed.
Point 4:
- Ways to improve thermal conductivity
There are many factors mentioned in this section. These issues could be synthesized (employed process, utilized polymer/copolymer, filler, outputs) and summarized in table form.
Response 4:
Thank you for your suggestions on our manuscript. Your comments were very useful to us. In response to this comment, we have collated the literature according to the method used and the purpose, matrix, fillers, and thermal conductivity achieved. And added "Table 3 summarizes the methods to improve the thermal conductivity of FFF 3D prints, followed by a brief introduction of related studies." to page 21, lines 663-666. The inserted table is at page 21 and 22.
We appreciate for Reviewer's warm work earnestly and hope that the correction will meet with approval. Once again thank you very much for your comments and suggestions.
Yours sincerely
Zewei Cai

Reviewer 2 Report
Please do not consider thermoplastic mateiral as curable, and change the language in line 239! "the print can be cured quickly...." is not correct.
you mention in Table 2 several materials, yet only a few are discussed later on. (Expandable Graphite, Fem CNTs are missing a paragraph) ideally this could be added or at least a line should be added that they are similar to ...
under 6.2 the term GNP is not explained and please link it in a way to table 2 as it is obviously Graphene Nano platelets.
Author Response
Response to Reviewer 2 Comments
Dear Reviewer:
Thank you for your valuable comments on our paper entitled " Progress of Polymer-Based Thermal Conductive Materials by Fused Filament Fabrication: A Comprehensive Review" (Manuscript ID: polymers-1941164). Those comments are all valuable and very helpful for revising and improving our paper. We have studied comments carefully and have made correction which we hope meet with approval. The corrections are marked in red in the manuscript. The major revisions in the paper and responses to your comments are listed below.
Comment 1:
Please do not consider thermoplastic mateiral as curable, and change the language in line 239! "the print can be cured quickly...." is not correct.
Response 1: Thank you very much for your valuable comments on the manuscript. The article incorrectly describes the thermoplastic material partly as curable. We made two changes to address this error. The first is on page 2, lines 56-58, where the basic principles of the different 3D printing technologies were originally described. This includes FFF technology, which uses thermoplastic materials for printing, so the term "cured" is inappropriate. I changed it to "Then, the liquid or powder material is bonded layer by layer to form the final prints" to replace the original expression." The second change is on page 9, lines 268-272. It was intended to express the advantages of FFF technology of rapid prototyping and no complex post-processing but was incorrectly described as "cured." We have changed this place to "Lastly, the prints produced by FFF can be formed quickly, eliminating the need for lengthy and complex post-processing for prints that do not require a support structure."
Comment 2: you mention in Table 2 several materials, yet only a few are discussed later on. (Expandable Graphite, Fem CNTs are missing a paragraph) ideally this could be added or at least a line should be added that they are similar to ...
Response 2: We greatly appreciate the reviewers' suggestions for the manuscript. Many fillers are mentioned in Table 2, while only some are introduced next, missing expandable graphite, Fe, and carbon nanotubes, resulting in a lack of rigor in the article. To address this issue, I have added descriptions of several similar fillers. Since the fillers are relatively similar, only the more studied and representative fillers are selected as examples for a detailed introduction. Added the following to page 13, lines 361-371: "As shown in Table 2, the FFF thermally conductive fillers mainly include graphene, expandable graphite, carbon fiber, carbon nanotubes, BN, AIN, Cu, Fe, SiC, etc. Expandable graphite can be used as a precursor for preparing graphene, as opposed to researchers who are more interested in using graphene to improve the thermal conductivity of printed parts. Carbon fibers and nanotubes are one-dimensional carbon fiber materials with similar properties, so the more used carbon fiber was chosen as a representative. Cu and Fe are metal fillers, and both have similar thermal conductivity mechanisms, so Cu, which has more relevant studies and is more representative, is chosen for the introduction. The fillers with similar properties are no longer described in detail. They are divided into two main categories according to the electrical properties of thermally conductive fillers in the following part."
Comment 3: under 6.2 the term GNP is not explained and please link it in a way to table 2 as it is obviously Graphene Nano platelets.
Response 3: Thank you for your detailed reading of the manuscript and valuable comments. I rechecked the original manuscript carefully and found that GNPs first appear on page 12, line 333 of 4.1.1, "graphene nanoplatelets (GNPs) can observe the agglomeration phenomenon." Since the abbreviation GNPs was explained here, the GNPs are not described further when they appear again in 6.2. The use of "graphene" in some parts of Table 2 is rather broad and does not link the table's contents to the contents of 6.2. To address this problem, I rechecked the wording in the reference and changed it to the specific words in the article. The specific changes are reflected in the names of the fillers of references 64, 66, 69, 72, and 73 in Table 2. Since the GNPs have been described in an expanded manner in the table, the abbreviations are used instead after that.
We appreciate for Reviewer's warm work earnestly and hope that the correction will meet with approval. Once again thank you very much for your comments and suggestions.
Yours sincerely
Zewei Cai
